# AtFAHD1a: A New Player Influencing Seed Longevity and Dormancy in Arabidopsis?

**DOI:** 10.3390/ijms22062997

**Published:** 2021-03-15

**Authors:** Davide Gerna, Erwann Arc, Max Holzknecht, Thomas Roach, Pidder Jansen-Dürr, Alexander K.H. Weiss, Ilse Kranner

**Affiliations:** 1Department of Botany, University of Innsbruck, Sternwartestraße 15, 6020 Innsbruck, Austria; erwann.arc@uibk.ac.at (E.A.); thomas.roach@uibk.ac.at (T.R.); ilse.kranner@uibk.ac.at (I.K.); 2Center for Molecular Biosciences Innsbruck (CMBI), University of Innsbruck, 6020 Innsbruck, Austria; max.holzknecht@uibk.ac.at (M.H.); pidder.jansen-duerr@uibk.ac.at (P.J.-D.); 3Research Institute for Biomedical Aging Research, University of Innsbruck, Rennweg 10, 6020 Innsbruck, Austria

**Keywords:** antioxidants, ascorbate, FAH superfamily, fumarylacetoacetate hydrolase, glutathione, mitochondria, seed ageing, seed development, seed dormancy, seed longevity

## Abstract

Fumarylacetoacetate hydrolase (FAH) proteins form a superfamily found in Archaea, Bacteria, and Eukaryota. However, few fumarylacetoacetate hydrolase domain (FAHD)-containing proteins have been studied in Metazoa and their role in plants remains elusive. Sequence alignments revealed high homology between two *Arabidopsis thaliana* FAHD-containing proteins and human FAHD1 (hFAHD1) implicated in mitochondrial dysfunction-associated senescence. Transcripts of the closest hFAHD1 orthologue in Arabidopsis (AtFAHD1a) peak during seed maturation drying, which influences seed longevity and dormancy. Here, a homology study was conducted to assess if AtFAHD1a contributes to seed longevity and vigour. We found that an *A. thaliana* T-DNA insertional line (*Atfahd1a-1*) had extended seed longevity and shallower thermo-dormancy. Compared to the wild type, metabolite profiling of dry *Atfahd1a-1* seeds showed that the concentrations of several amino acids, some reducing monosaccharides, and δ-tocopherol dropped, whereas the concentrations of dehydroascorbate, its catabolic intermediate threonic acid, and ascorbate accumulated. Furthermore, the redox state of the glutathione disulphide/glutathione couple shifted towards a more reducing state in dry mature *Atfahd1a-1* seeds, suggesting that AtFAHD1a affects antioxidant redox poise during seed development. In summary, AtFAHD1a appears to be involved in seed redox regulation and to affect seed quality traits such as seed thermo-dormancy and longevity.

## 1. Introduction

Members of the fumarylacetoacetate hydrolase (FAH) superfamily are found in Archaea, Bacteria, and Eukaryota, displaying wide substrate specificity [1,2,3]. Surprisingly, despite a conserved FAH domain (FAHD) and high structural similarities, FAHD-containing enzymes catalyse distinct reactions, including hydrolysis of ß-diketones, decarboxylation, and isomerisation [3,4,5,6,7]. Nonetheless, FAHD-containing proteins remain poorly explored, especially in plants. Proteomics studies comparing young and senescent human cell cultures revealed posttranslational modifications of a FAHD-containing protein, identified as human FAHD1 (hereafter referred to as hFAHD1), pointing at its involvement in the control of cellular senescence [8,9]. hFAHD1 shows high homology to two other human proteins named hFAHD2a and hFAHD2b, whose function remains yet unexplored [10,11]. Conversely, due to its mitochondrial localisation and displaying both acylpyruvate hydrolase (EC 3.7.1.5) and oxaloacetate decarboxylase (EC 4.1.1.112) activity, hFAHD1 is proposed to act antagonistically to the anaplerotic reaction catalysed by pyruvate carboxylase in the metabolic flux of the tricarboxylic acid (TCA) cycle [5,6,11,12,13]. Furthermore, the depletion of hFAHD1 inhibits mitochondrial metabolism, suggesting a role in mitochondrial dysfunction associated with senescence of human cells [8,14]. In mammals, ageing has been related to an accumulation of senescent cells, which progressively lose their replicative potential due to persistent withdrawal from the cell cycle [15,16]. In the model invertebrate *Caenorhabditis elegans,* knocking down the hFAHD1 orthologue impairs mitochondrial function, locomotion, egg-laying behaviour, and neuronal transmission [7,17]. However, little is known about FAH superfamily proteins in plants.

The genome of the model plant *Arabidopsis thaliana* contains three genes, located on different chromosomes, encoding putative FAHD-containing proteins: *AT1G12050*, *AT3G16700*, and *AT4G15940*, among which only the form encoded by *AT1G12050*, referred to as fumarylacetoacetate hydrolase (AtFAH), has as yet received attention. *In vitro* studies show its ability to hydrolyse fumarylacetoacetate, the last step committed to tyrosine catabolism, which is relevant to survive dark-induced senescence and cell death [18,19,20,21,22]. Declining seed viability should not be confused with plant senescence, which is defined as a genetically controlled developmental phase that requires nutrient mobilisation mediated by sugar signalling and hormonal regulation [23]. Seed ageing usually entails changes in the embryo and other seed structures over time, starting early upon seed development and maturation and ultimately leading to seed death [23,24]. Over the whole life cycle of *A. thaliana*, *AT4G15940* transcripts accumulate the most in seeds during late maturation and desiccation [25,26] (https://bar.utoronto.ca/efp/cgi-bin/efpWeb.cgi accessed on 15 January 2021), a period during which the key seed traits of dormancy (i.e., a block to the completion of germination of an intact viable seed placed under-temporary-favourable conditions in an otherwise unfavourable season) and longevity are critically influenced [27,28,29,30].

Most spermatophytes, including *A. thaliana*, produce desiccation tolerant, so-called “orthodox” seeds. In the final stage of seed maturation, seeds desiccate, and their water content (WC) enters into equilibrium with environmental humidity [31,32]. Below an average WC threshold of about 0.1 g H_2_O g^−1^ dry weight (DW), seeds form a glassy (i.e., highly viscous) cytoplasm, whereby molecular relaxation, diffusion, and metabolic reactions are severely restricted, imposing a state of quiescence [33,34,35]. Due to its influence on metabolic rates of desiccated seeds, seed WC is negatively correlated with seed longevity and thus storability [36]. Therefore, it is possible to alter seed ageing rates, making seeds convenient models to study mechanisms of ageing and viability loss. Seeds can be rapidly aged by exposure to elevated temperatures (35–45 °C) and relative humidity (RH > 60%) typical of controlled deterioration (CD) protocols [37]. Another method of artificial ageing is “accelerated ageing” (AA), a treatment in which seeds are aged in a water-saturated atmosphere approximating 100% RH at constant temperature (International Seed Testing Association, 2018 [38]). The loss of seed viability has been associated with oxidative modifications within seeds [39,40,41,42,43,44,45,46,47,48], in line with the “free radical theory of ageing” (FRT), which postulates that the accumulation of damage over the life-time of an organism induced by free radicals and reactive oxygen species (ROS) impairs viability [49]. Whereas the original concept of the FRT was very instructive to develop a conceptual framework of how stress and molecular damage can contribute to ageing, more recent versions of the FRT assume that "free radicals and related oxidants are but one subset of stressors with which all life forms must cope over their lifespans" [50]. Nonetheless, desiccated orthodox seeds are exposed to ROS-induced molecular damage, as they cannot prevent ROS formation in an oxygenic environment, and the activity of ROS-processing enzymes is severely restricted in the glassy state [46,51,52,53]. Thus, seed protection from oxidative macromolecular damage requires the presence of antioxidants.

Seeds are equipped with a set of antioxidants located in both the aqueous (i.e., cytoplasm) and lipid (i.e., membranes and lipid bodies) domains. The most abundant water-soluble antioxidant and redox buffer in mature dry orthodox seeds is the low-molecular-weight (LMW) thiol glutathione (γ-l-glutamyl-l-cysteinyl-glycine, GSH), followed by l-ascorbic acid (l-threo-hexenon-1,4-lactone or vitamin C, AsA), usually present at much lower or hardly detectable concentrations [43,54,55,56]. GSH and AsA can directly scavenge ROS or act as electron donors to ROS-processing enzymes [57], such as glutathione-*S*-transferase (EC 2.5.1.18) and ascorbate peroxidase (EC 1.11.1.11). This results in a temporary accumulation of glutathione disulphide (GSSG) and a small fraction of l-dehydroascorbic acid (DHA), whose regeneration and *de novo* synthesis can resume only with seed water uptake during germination [58,59,60,61,62]. In cell membranes and storage bodies, the main lipophilic seed antioxidants belong to the “vitamin E” or “tocochromanols family” comprising tocopherols, tocotrienols, and tocomonoenols, which can scavenge lipid peroxides [63,64,65]. The tocopheryl radicals thereby produced are regenerated at the expense of AsA and GSH [66,67]. Therefore, the availability of reduced forms of antioxidants is critical to maintaining ROS levels within an “oxidative window” that ensures cellular signalling functions rather than deteriorative reactions at the basis of seed ageing [68,69,70]

Here, we hypothesised that hFAHD1 orthologues in *A. thaliana* have a conserved physiological function related to cell longevity. We first conducted bioinformatics analyses to identify the closest orthologue to hFAHD1, whose transcripts are predicted to be expressed at highest levels in mature dry seeds. We then characterised seed longevity and dormancy phenotypes in two successive generations of a wild type (WT) and a *FAHD1* (*AT4G15940*) T-DNA insertional mutant line of *A. thaliana*. Metabolic and redox profiles of dry seeds were measured to investigate biochemical processes involving the hFAHD1 plant orthologue, which could be related to seed longevity. This is the first study on FAHD-containing proteins in plants, and it shows that the closest orthologue to hFAHD1 appears to play a role in seed longevity. Our results also extends current knowledge on the FAH superfamily across Eukaryota.

## 2. Results

### 2.1. Sequence Alignments Indicated Structural Homologies between Two Arabidopsis FAHD-Containing Protein Isoforms and hFAHD1

Blasting the amino acid sequences of the proteins encoded by *AT4G15940* and *AT3G16700* against hFAHD1 revealed sequence identities of 53 and 49%, respectively, including conserved residues and motifs (Figure 1). Therefore, *AT4G15940* and *AT3G16700* are hereafter referred to as *AtFAHD1a* and *AtFAHD1b*, respectively, and the corresponding proteins as AtFAHD1a and AtFAHD1b, in line with the nomenclature assigned to hFAHD2a and hFAHD2b. However, sequence alignments also showed that AtFAHD1a and AtFAHD1b differ from FAHD2 proteins, including those of rat and mouse, in their N-terminus (Figure 1). Only one transcript is encoded by *AtFAHD1a*, whereas four splice variants were identified for AtFAHD1b (Appendix A). In contrast, *AT1G12050* is orthologue to the human FAH (hFAH; Figure 1) and has been already experimentally confirmed as FAH enzyme in Arabidopsis (AtFAH), with highest expression levels in senescing leaves and flower buds [18,19,20,21,22] (Appendix A). 

Publicly available microarray data (eFP Arabidopsis browser; https://bar.utoronto.ca/efp/cgi-bin/efpWeb.cgi accessed on 15 January 2021) for different plant organs and at various growth stages showed maximal expression of *AtFAHD1a* in the embryo of fully mature and desiccated seeds (Appendix A), whereas the expression pattern of *AtFAHD1b* was more heterogeneous, with relatively low expression in rosette leaves, shoot apex, and immature seeds (Appendix A). Therefore, considering its expression pattern in seeds and a slightly higher homology to hFAHD1, AtFAHD1a was chosen for experiments to reveal a seed phenotype. Based on the subcellular localisation database for Arabidopsis protein (http://bar.utoronto.ca/cell_efp/cgi-bin/cell_efp.cgi accessed on 15 January 2021; [71]), the locations of AtFAHD1a and AtFAHD1b were predicted to be predominantly mitochondrial, similar to their closest orthologues in human. Conversely, AtFAH was predicted to be mostly cytoplasmic, as supported by available studies on this FAH superfamily member.

### 2.2. Atfahd1a-1 Seeds Had a Less Thermo-Dormant Phenotype

Immediately after harvest, Arabidopsis seeds can exhibit thermo-dormancy. Thermo-dormancy is defined as a particular type of primary dormancy imposed by temperature, which enables the germination of dormant seeds within a narrower range of temperature than non-dormant seeds. Germination tests at 25 °C, a temperature preventing germination of thermo-dormant seeds, showed that mutant seeds (hereafter referred to as *Atfahd1a-1*) of two plant generations could reach a total germination (TG) up to 40% higher than the WT (Figure 2A,C). Furthermore, at 25 °C *Atfahd1a-1* seeds reached 50% TG within 2 d, which was about 17 h earlier than WT seeds across both generations (Figure 2B,D). At 20 °C, a temperature less restrictive to germination of thermo-dormant seeds, TG of both genotypes was close to 100%, but *Atfahd1a-1* seeds still germinated faster than WT seeds across the first (F_1_) and the second plant generation (F_2_) (Figure 2). For Arabidopsis, a temperature of 15 °C is more permissive for germination of thermo-dormant seeds than 20 °C. Accordingly, seeds of both lines (only tested in seeds from F_2_ plants) achieved maximal TG, but without cold stratification *Atfahd1a-1* seeds germinated faster than WT seeds (Figure 2B,D). After cold stratification, the germination rates of both *Atfahd1a-1* and WT seeds did not differ at 20 °C (Figure 2B,D). Seeds produced by F_2_ plants were also used to investigate the sensitivity of germination to abscisic acid (ABA), whereby endosperm rupture after 10 d was significantly inhibited only at 1 and 5 µM ABA (*p* = 0.048 and 0.007, respectively) compared to seeds germinated without ABA (Appendix A). Nonetheless, differences between the two lines were not significant for all tested ABA concentrations (Appendix A). 

In summary, *Atfahd1a-1* seeds reached higher TG at 25 °C and germinated faster than the WT under all tested temperatures across two successive plant generations without cold stratification, which, in turn, removed these differences.

### 2.3. Atfahd1a-1 Seeds Showed a Longevity Phenotype after CD and Ageing under Ambient Temperatures

To assess if AtFAHD1a affects seed longevity, seeds of both genotypes were subjected to CD at 40 °C and RHs of 59.4 ± 1.1% and 74.5 ± 0.5%, as well as to AA at 100% RH. Seeds lost viability faster at the highest RHs, as indicated by seed-survival curves and time to decrease seed viability by 50% (referred to as P50) (Figure 3; Appendix A). At 59.4 ± 1.1% RH, *Atfahd1a-1* seeds retained higher TG than WT seeds after 49 days of CD (Figure 3A). After 77 d, the TG of WT seeds significantly dropped to about 60%, whereas it was, on average, 26% higher in *Atfahd1a-1* seeds (Figure 3A). At 74.5 ± 0.5% RH, *Atfahd1a*-1 seeds retained 20% higher TG than WT seeds after 16 d of CD. However, seed viability of both genotypes was drastically lost within the next 6 d (Figure 3B), overall resulting in significantly lower P50 in WT seeds (Figure 2B; Appendix A). At 100% RH, *Atfahd1a-1* seeds germinated by only 6% more than WT seeds within 2 d after AA (*p* > 0.05). However, only a quarter of the seeds from either genotype had survived after 8 d (Figure 2C), and P50 values did not differ significantly (Figure 3C; Appendix A). Similar to CD at 59.4 ± 1.1% RH, *Atfahd1a-1* seeds also maintained higher TG than WT seeds after 4 and 8 months of ageing under ambient temperatures of 24.5 ± 2.0 °C and 68.2 ± 2.0% RH (Figure 3D; Appendix A). 

In summary, compared to the WT, *Atfahd1a-1* seeds retained extended longevity both after ageing under ambient temperatures and CD at 40 °C and RH ≤ 75%, conditions that corresponded to seed WCs ≤ 0.1 g H_2_O g^−1^ DW. In contrast, exposing seeds to AA at 100% RH, under which seeds increased their WC to 0.22 g H_2_O g^−1^ DW, did not reveal a distinguishable *Atfahd1a-1* phenotype (Appendix A).

Considering its predicted mitochondrial subcellular localisation, the possible contribution of AtFAHD1a to mitochondrial function was investigated by measuring oxygen consumption rates (OCRs) during the first hours of imbibition at 21 °C. In *Atfahd1a-1* seeds, maximal respiration, obtained by subtracting the baseline levels from the peak of O_2_ consumption, was 1.3-fold higher than in WT seeds (Figure 4A; Appendix A). Following the addition of KCN, an inhibitor of complex IV (i.e., cytochrome *c* oxidase) of the mitochondrial electron transport chain (ETC), O_2_ consumption dropped to the initial basal levels in both lines, revealing that it was mostly related to mitochondrial respiration, and that the activity of complex IV was significantly higher in *Atfahd1a-1* seeds (Figure 4B). Seed exposure to *n*-octyl gallate, a well-established inhibitor of plant mitochondrial alternative oxidase [72], did not significantly further decrease OCR in either line (Appendix A). 

In summary, *Atfahd1a-1* seeds displayed a faster resumption of OCR than WT seeds following the onset of imbibition.

### 2.4. Atfahd1a-1 Seeds Maintained Higher Viability after CD at 60% RH and Dormancy-Breaking Treatments

To clarify if the *Atfahd1a-1* seed germination phenotype after ageing was indicative of differences in longevity and to exclude the presence of residual dormancy, F_2_ seeds were subjected to CD at 40 °C and 59.4 ± 1.1% RH for 77 d, after which their viability significantly decreased (Figure 2A). Thereafter, *Atfahd1a-1* seeds incubated at 20 °C reached 27% higher TG than WT seeds and germinated more rapidly, regardless of cold stratification (Figure 5). The availability of exogenous gibberellic acid (GA_3_) during seed germination did not attenuate the differences in TG at 20 °C between the two genotypes (Figure 5). Moreover, the TG of *Atfahd1a-1* seeds subjected to CD and germinated at 15 °C remained by 19% significantly higher than in WT seeds (Figure 5). 

Overall, the higher TG of aged *Atfahd1a-1* seeds relative to WT seeds was due to an extended longevity, rather than residual dormancy in WT seeds.

### 2.5. Contents of Low-Molecular-Weight Metabolites and Antioxidants were Affected in Atfahd1a-1 Seeds 

In non-aged dry seeds, 99 LMW metabolites were identified, 21 of which differed significantly in abundance between *Atfahd1a-1* and WT seeds (Appendix A; Figure 6). Among the differentially accumulated LMW seed metabolites, 18 were less abundant and three accumulated in *Atfahd1a-1* seeds compared to WT seeds (Figure 6). Amino acids represented the largest biochemical class distinguishing the metabolite profile of the two genotypes, whereby the abundance of several proteinogenic amino acids and the non-proteinogenic γ-aminobutyric acid (GABA) were lower in *Atfahd1a-1* seeds. The pentose arabinose, the hexose ribose, and the deoxy sugar rhamnose followed a similar trend (Figure 6). Other metabolites with lower levels in *Atfahd1a-1* seeds than in WT included the polyols glycerol and its phosphorylated form glycerol-3-phosphate, linoleic acid, hydroxylamine, the nucleobase uracil, and pipecolic acid (Figure 6). Notably, compared to WT seeds, *Atfahd1a-1* seeds had higher levels of the water-soluble antioxidant AsA (2.8-fold), its corresponding oxidised form dehydroascorbate (DHA, 2.0-fold), and the DHA break-down product threonic acid (1.3-fold). In contrast, the lipid-soluble antioxidant δ-tocopherol became 0.7-fold less abundant (Figure 6).

A targeted reversed-phase high performance liquid chromatography (HPLC) method was chosen to determine the concentrations and redox state of LMW thiols, including the antioxidant GSH and its metabolic intermediates cysteine (cys), γ-l-glutamyl-cysteine (γ-glu-cys), and cysteinyl-glycine (cys-gly). Total glutathione (i.e., GSH + GSSG) was the most abundant LMW thiol-disulphide redox couple in seeds of both lines, exceeding by one order of magnitude the contents of the other LMW thiols (Figure 7A,D–F). Despite equivalent total glutathione concentrations, *Atfahd1a-1* seeds contained 15% less GSSG (Figure 7A). Therefore, less GSH was oxidised to GSSG in *Atfahd1a-1* seeds, indicative of a more reducing thiol-based cellular redox state, which was confirmed by a 13 mV less negative half-cell reduction potential of the GSSG/2GSH redox couple (E_GSSG/2GSH_) than in WT seeds (Figure 7B). Seed WCs, used to calculate E_GSSG/2GSH_ through the molar concentrations of GSH and GSSG, did not significantly differ between the two genotypes (Figure 7C). The comparisons of cys, γ-glu-cys, cys-gly concentrations, together with those of their corresponding disulphides, revealed no significant differences between the two seed lines (Figure 7D–F).

Altogether, *Atfahd1a-1* seeds contained lower concentrations of several amino acids and a few reducing sugars, while showing altered redox profiles, characterised by more AsA and DHA, and less GSSG and δ-tocopherol.

## 3. Discussion

### 3.1. The FAH Superfamily in Arabidopsis and a Specific Role for AtFAHD1a in Seeds

In this paper, we contribute new knowledge on the functional role of an uncharacterised member of the FAH superfamily, named AtFAHD1a, in seeds of the model plant Arabidopsis. The proteins AtFAHD1a and AtFAHD1b were identified as alternative versions homologue to hFAHD1 (Figure 1), with *AtFAHD1a* and *AtFAHD1b* showing distinct expression patterns in the plant life cycle (Appendix A). This feature is common to other FAHD-containing proteins, such as hFAHD2a and hFAHD2b, which are predicted to be differently expressed in human organs and cell types [10] (Human Protein Atlas; http://www.proteinatlas.org accessed on 15 January 2021). In particular, expression levels of *AtFAHD1a* rise during seed development and reach a peak in fully mature dry seeds, before dropping during the first hours of germination (Appendix A) [26,73]. Recently, hFAHD1 has been characterised as a mitochondrial bi-functional enzyme, with acylpyruvase and oxaloacetate decarboxylase activities implicated in mitochondrial ETC and cell longevity [8,11,13]. The presence of conserved residues in the FAHD of AtFAHD1a suggested a potentially similar mechanism of action (Figure 1). Thus, based on sequence homology, we assessed if AtFAHD1a could also influence seed longevity and related seed-quality traits.

### 3.2. A Role of AtFAHD1a in Influencing Seed Thermo-Dormancy 

Seed longevity and the depth of seed dormancy are determined during late seed maturation, the phase of plant life cycle during which *AtFAHD1a* is most highly expressed (Appendix A) [26], and ultimately affects seed vigour [28,74,75]. The mutant line characterised in this study (i.e., *Atfahd1a-1*) contained a T-DNA insertion in the last exon, inducing a missense mutation, which likely resulted in a non-functional protein (PolyAllele Detail (arabidopsis.org); https://seqviewer.arabidopsis.org/servlets/sv?action=seq&start=9039904&length=10000&band=0&option2=6 accessed on 15 January 2021).

In seeds, temperature is a key environmental cue for germination and a regulator of dormancy level during late seed maturation [28,30,76]. During cold stratification, imbibed seeds become less dormant due to a change in the ratio between ABA and gibberellins [30,77,78]. Notably, *Atfahd1a-1* seeds revealed a shallower thermo-dormant phenotype than WT seeds, whereby significantly more non-stratified *Atfahd1a-1* seeds germinated at 25 °C after harvest. As typical of weakly dormant winter annual species, dormancy was less expressed in both genotypes at lower germination temperatures (e.g., 15 °C), leading to >95% TG. Nonetheless, *Atfahd1a-1* seeds still germinated faster than WT seeds at all tested temperatures (Figure 2), indicative of higher seed vigour. Early upon imbibition, higher OCRs were detected in *Atfahd1a-1* seeds, in line with their faster germination (Figure 4). Importantly, after cold stratification to relieve dormancy, the two seed lines displayed equivalent germination rates (Figure 2), supporting that the germination phenotypes observed before stratification were indeed dormancy-related. However, the influence of AtFAHD1a on seed germination appeared to be independent of ABA sensitivity (Appendix A).

### 3.3. Influence of AtFAHD1 on Seed Longevity in Relation to Cytoplasmic Physical State during Ageing

The higher viability of *Atfahd1a-1* seeds after ageing (Figure 3; Appendix A) draws attention to a possible role of AtFAHD1a in seed longevity. Importantly, differences in germination after ageing were not due to residual dormancy in WT seeds (Figure 5). Changes of the physical state of the cytoplasm can influence the nature of biochemical reactions occurring in seeds, and hence their longevity [39,51,79]. Notably, the longevity phenotype of *Atfahd1a-1* seeds was revealed after ageing under ambient temperatures and CD at 40 °C and 59.4 ± 1.1 and 74.5 ± 0.5% RH, but not after AA at 100% (Figure 3; Appendix A). Phase diagrams of various seed species described changes in cytoplasmic viscosity and molecular mobility as a function of WC and storage temperature [29,33,35]. According to seed phase diagrams constructed for *Pinus densiflora* with similar oil content of ca. 30% (*w*/*w*) compared to Arabidopsis Col-0 accessions [80], seeds exposed to CD at 40 °C and 59.4 ± 1.1 or 74.5 ± 0.5% RH (WC ~0.075 and ~0.099 g H_2_O g^−1^ DW, respectively) would possess a fluid-rubbery cytoplasm [51]. Similarly, seeds aged under ambient temperatures (WC ~0.086 g H_2_O g^−1^ DW) were likely in the same physical state, whereby the cytoplasm retained similar viscosity (Appendix A). However, at 100% RH, seeds reached a WC of about 0.220 g H_2_O g^−1^ DW, and cytoplasmic solutes presumably dissolved, further lowering cellular viscosity in comparison to lower RHs (Appendix A). Therefore, seed longevity was extended in the absence of a functional AtFAHD1a in the fluid-rubbery state, under which only restricted enzymatic activities are possible but may include redox-associated defence processes, such as the antioxidant functions of GSH and tocochromanols [51,64]. In contrast, AtFAHD1a did not significantly affect longevity in seeds with a less viscous cytoplasm (at 100% RH) (Figure 3; Appendix A), which enables a complete resumption of molecular mobility, diffusion, and hence enzyme activity [51,81,82]. 

To summarise, the extended seed longevity phenotype of *Atfahd1a-1* seeds, which was revealed at lower seed WCs, resulting in a fluid-rubbery cytoplasm more restrictive of metabolism, points to differences influenced by AtFAHD1a during seed development, rather than defence-related metabolism during ageing. 

Seed development includes multiple stages (i.e., morphogenesis, reserve accumulation, and desiccation) and is accompanied by extensive metabolic rearrangements [31,83]. Therefore, to gain insights into the consequences of *Atfahd1a-1* knockout on seed-quality traits, profiles of primary metabolites were characterised in mature dry seeds of both lines. Notably, *Atfahd1a-1* seeds contained lower concentrations of several free amino acids (Figure 6). Mobilisation and *de novo* synthesis of free amino acids prevail their incorporation into storage proteins during late seed maturation [84]. Particularly, free amino acid accumulation is typical of seed desiccation, which prepares seeds for germination by promptly providing precursors and metabolites ready-to-use upon imbibition [31,55,83]. For instance, branch-chained amino acids (i.e., valine, leucine, and isoleucine) and the non-proteinogenic amino acid GABA can fuel the TCA cycle during the first hours of germination [85,86]). *Atfahd1a-1* seeds showed drops in isoleucine, leucine, and GABA concentrations, accompanied by a decline in alanine and β-alanine (Figure 6), which can be generated as by-products of mitochondrial GABA metabolism [86]. However, as *Atfahd1a-1* seeds resumed maximal aerobic respiration earlier and germinated faster than the WT (Figure 4), the lowered amino acid concentrations in *Atfahd1a-1* seeds did not apparently restrict the onset of germination. Furthermore, lysine can be a precursor of pipecolic acid, a defence compound implicated in the induction of systemic acquired resistance to biotic stress [87]. As the levels of both lysine and pipecolic acid dropped (Figure 6), lysine metabolism was potentially affected in *Atfahd1a-1* seeds. 

Apart from amino acids, lower concentrations of reducing monosaccharides (i.e., arabinose, rhamnose, and ribose) and δ-tocopherol (Figure 6), which accumulate during seed desiccation [28,83], may have influenced the longevity of *Atfahd1-1a* seeds. Firstly, reducing sugars are implicated in non-enzymatic reactions with amino acids, forming Amadori and Maillard products (i.e., glycated proteins), which negatively correlate with longevity [28,88,89]. Secondly, knockout mutants have shown that tocochromanols contribute to prevent seed deterioration and protect seeds from lipid peroxidation, following desiccation [90,91]. In Arabidopsis Col-0 seeds, γ-tocopherol is the most abundant form, followed by δ- and α- tocopherols [90]. Compared to the WT, only δ-tocopherol was significantly lower in *Atfahd1a-1* seeds (Appendix A). Therefore, lower concentrations of reducing monosaccharides would agree with the extended longevity of *Atfahd1a-1* seeds, whereas the decline of tocopherols would apparently not support it. However, previous analyses have shown that the tocochromanol pool remains stable in response to seed ageing by CD under conditions similar to those used in this study [44,48,64], indicating that a limited difference in a minor tocopherol form is likely not impairing seed longevity. 

In summary, knocking out *AtFAHD1a* resulted in dry seeds displaying altered contents of some LMW metabolites, which are implicated in the remodelling of seed metabolome during seed late maturation and desiccation.

### 3.4. Enhanced Antioxidant Defences and Extended Longevity of Atfahd1a-1 Seeds 

During embryogenesis of orthodox seeds, embryos rely on the parental plant for AsA supply and become equipped with the enzymes required for autonomous AsA synthesis and regeneration right before desiccation [54,92,93]. Thereupon, AsA contents drastically decrease, potentially because of pro-oxidant activity of AsA. Molecular crowding in the desiccated state may force AsA to promote Fenton reactions in the presence of transition metals [54,94,95,96]. Among all LMW metabolites identified by the GC-MS-based metabolite profiling, only AsA, DHA, and threonic acid became significantly more abundant in *Atfahd1a-1* seeds compared to the WT (Figure 6; Appendix A). Threonic acid can be produced during DHA catabolism [97,98] and, in conjunction with elevated concentrations of AsA and DHA, points to a possible interference of AtFAHD1a with AsA metabolism during late seed maturation.

In contrast to AsA, autonomous GSH synthesis is already active in early developing embryos and remains essential for seed maturation and desiccation [54,99]. Furthermore, the contribution of GSH to seed longevity has been much more extensively reported [43,58,100]. On the one hand, shifts in the redox state of glutathione towards more oxidising conditions have been correlated with loss of seed viability after CD [40,41,43,45,46,101], while, on the other hand, tissue-dependent thiol-disulphide redox shifts also accompany seed germination and seedling growth [55,60,102]. Notably, dry *Atfahd1a-1* seeds had more reducing cellular conditions (i.e., more negative E_GSSG/2GSH_ values) (Figure 7), which agrees with both their longevity and germination phenotypes. The influence of AtFAHD1 on longevity was more prominent when seeds were aged in the fluid-rubbery state (Figure 3; Appendix A), under which seeds have very limited metabolism and rely on LMW antioxidants for protection from oxidative damage [39,100,103]. Concerning seed germination, reduction of disulphides can occur within minutes to hours of seed water uptake [60,104], which is important in resuming mitochondrial metabolism, including the TCA cycle [105,106]. Thus, an initially more reducing cellular environment may contribute to faster germination of *Atfahd1a-1*, a relationship previously shown in wheat seeds [55]. 

In summary, AtFAHD1a appeared to participate in seed redox regulation during seed late maturation and desiccation.

## 4. Materials and Methods

### 4.1. Plant Material and Seed Production

WT seeds of *Arabidopsis thaliana* L. (thale cress or Arabidopsis) of the ecotype Columbia (Col-0) were compared with an insertion line of the SALK collection in the Col-0 background (SALK_020157C, [107]), both provided by the Nottingham Arabidopsis Stock Centre (United Kingdom) and used as parental generation to produce two generations of pure seed lines. The mutant line had a T-DNA insertion of 390 bp (between 9039914 and 9040304 bp) in the last of seven exons of the *AT4G15940* gene, 300 bp before the 3’ end (https://www.arabidopsis.org/ accessed on 15 January 2021; https://seqviewer.arabidopsis.org/?action=accession&type=tdna&id=504961276&chr=4 accessed on 15 January 2021; https://seqviewer.arabidopsis.org/servlets/sv?action=seq&start=9036904&length=10000&band=0&option2=5 accessed on 15 January 2021). Five seeds per pot (*n* = 10–15) were cold stratified for 4 d at 4 °C in wet commercial soil mix (*Einheitserde Classic*, Einheitserdewerke Patzer PATZER ERDEN GmbH, Germany), before transfer to controlled growth conditions (20 °C day/16 °C night cycle, with 16 h of light at 99 ± 13 μmol photons m^−2^ s^−1^) for about 13 weeks, required for the production of fully mature seeds. At the rosette stage, plants were sheltered with Aratubes (*Arasystem*, BETATECH BVBA, Ghent, Belgium) to prevent cross-pollination between the two parental genotypes (i.e., *Atfahd1a-1* and WT) for the production of F_1_ seeds. Regular watering was stopped when first siliques senesced. F_1_ seeds produced by individual parental plants were harvested at full maturity before equilibration over a non-saturated LiCl solution (36–38% RH) for 6 d at room temperature (RT). For long-term storage, desiccated F_1_ seeds were hermetically sealed in a plastic box over silica gel and preserved at −20 °C until further analyses. 50 F_1_ seeds for each individual plant pool were germinated for three weeks in sealed Petri dishes over three layers of filter paper (Whatman grade 1, GE Healthcare, Little Chalfont, United Kingdom), and seedlings were used for DNA extraction and PCR screening of homozygous genotypes. All heterozygous F_1_ seeds were disregarded, and only F_1_ seeds obtained from F_1_ plants homozygous for the T-DNA insertion or the WT gene *AT4G15940* were used for both germination assays and production of a F_2_, grown as described for the F_1_. At full maturity, F_2_ seeds were also pooled from individual plants, grown for 3 weeks at the seedling stage and further screened for the homozygosity of both WT and *Atfahd1-1* lines. Only pure F_2_ seed lines were included in the germination assays to reveal phenotypes.

### 4.2. Characterisation of the T-DNA Insertion Line

Genomic DNA was extracted from typically 50 3-week-old seedlings (approximately 2 mg DW), using the cetyltrimethyl ammonium bromide method described by Clarke [108]. To confirm the homozygosity of the T-DNA insertions, paired PCR assays were conducted using *AT4G15940* specific primers (left primer 5′-ACATGAGGATCATATGCCCTG-3′ and right primer 5′-CTTGAGAGAGGATTTGAGCCC-3′), primers to amplify the T-DNA insertion of the mutant line (LBb1.3 5′-ATTTTGCCGATTTCGGAACCA-3′ and RP 5′-CTTGAGAGAGGATTTGAGCCC-3′), and a REDTaq^®^ ReadyMix^TM^ (Sigma-Aldrich, St. Louis, MO, USA). PCR conditions included an initial step of 5 min at 95 °C and 29 cycles consisting of 30 s at 95 °C, 40 s at 57 °C, and 1 min 40 s at 72 °C, followed by a final extension step of 10 min at 72 °C. The size of the PCR products was checked by agarose gel electrophoresis [109].

### 4.3. Seed Germination and CD

Following equilibration after harvesting, a minimum of six replicates of 100 seeds pooled from individual plants of both lines and generations were imbibed with 4 mL of ultrapure H_2_O on three layers of filter paper (diameter: 85 mm; Whatman grade 1, GE Healthcare, Little Chalfont, United Kingdom) in 9-cm Petri dishes sealed with Parafilm. Germination assays were conducted at 25 °C with 16 h of light at 49 ± 5 μmol photons m^2^ s^−1^ and 8 h of dark to investigate the occurrence of primary dormancy. The light intensity and photoperiod were maintained constant across all germination assays. A seed was regarded as germinated when the radicle tip protruded through the endosperm [110], and TG was scored under a SZ51 binocular stereomicroscope (Olympus Europa SE & Co. KG, Hamburg, Germany) at intervals up to 14 d, when non-germinated seeds showed first signs of decomposition. The rate of germination, an index of seed vigour, was calculated as time to reach 50% or 25% germination when TG was lower than 50% [111]. When mentioned, primary dormancy was released by cold stratification at 4 °C for 4 d in the dark before germination at 20 °C. To achieve maximal TG, seeds were germinated at 15 °C, a temperature that allows germination of thermo-dormant Arabidopsis seeds [112]. Furthermore, to determine their sensitivity to ABA, five replicates of 50 F_2_ seeds of both lines were sown on filter paper soaked with 4 mL of ultrapure H_2_O before 4 d of cold stratification at 4 °C. Thereafter, samples were dried-back under a laminar flow bench, imbibed with ABA solutions (0–5 µM range), and incubated at 25 °C. Seeds were scored at regular intervals for both testa and endosperm rupture, the latter of which is regulated by ABA [110,113].

To characterise seed longevity, seeds were equilibrated at RT in hermetically sealed plastic boxes over non-saturated LiCl solutions at 59.4 ± 1.1% and 74.5 ± 0.5% RH, measured with a hygrometer (HC2-AW-USB, Rotronic, Ettlingen, Switzerland) or at 100% RH using ultrapure H_2_O. Thereafter, seeds were incubated in the dark at 40 °C and the stability of temperature and moisture conditions over time were monitored with data loggers (*EasyLog*, Lascar Electronics Ltd., Whiteparish, United Kingdom). Seeds of both generations were tested at 59.4 ± 1.1% RH and only seeds of the F_1_ generation were subjected to other ageing regimes. To assess the effect of ageing under ambient temperatures on viability, seeds of the F_1_ generation were equilibrated and stored at indoor ambient temperatures (24.5 ± 2.0 °C, March 2018–February 2019) and 68.2 ± 2.0% RH. At intervals during the CD treatments, seeds (*n* = 6 replicates of 100 seeds for both lines) were incubated at 20 °C to assess their viability by scoring TG. Finally, after CD at 59.4 ± 1.1% RH for 11 weeks, the TG of F_2_ generation seeds was also assessed under the following conditions: (1) at 20 °C, (2) at 20 °C after 4 d of cold stratification at 4 °C, (3) at 20 °C after imbibition with 1 µM GA_3_, and (4) at 15 °C. Seed WCs during CD, AA, and ageing under ambient temperatures were estimated using water sorption isotherms as determined by Hay et al. [114]. 

### 4.4. Metabolite Extraction, Derivatisation, and GC-MS-Based Metabolite Profiling

Dry seeds of the F_1_ (*n* = 4 replicates of ca. 14 mg seed fresh weight (FW) pooled from individual plants of both lines) were frozen in liquid nitrogen and lyophilised for 5 d before storage over silica gel at −80 °C in a hermetically sealed plastic box until analysis. Right before extraction, samples cooled to −80 °C were ground with two 2-mm diameter agate beads using a bead mill (Tissue Lyser II, Qiagen, Hilden, Germany) set at 30 Hz for 2 min. Then, 13.39 ± 0.89 mg of finely ground seed powder were resuspended in 1 mL of pre-cooled (−20 °C) water:acetonitrile:isopropanol (2:3:3, v:v:v), containing 21.26 µM ^13^C_6_-sorbitol and 25 µM ^13^C_6_, ^15^N-valine (Sigma-Aldrich, St. Louis, MO, USA) as internal standards, and LMW metabolites were extracted for 10 min at 4 °C with continuous shaking at 1000 rpm (Compact Digital Micro plate shaker, Thermo Scientific, Waltham, MA, USA). Insoluble material was removed by centrifugation at 20,000× *g* for 5 min at 4 °C. Then, a 25-µL aliquot of the supernatant was collected and dried for 3 h in a vacuum centrifuge (Savant SPD111V P2 SpeedVac kit, Thermo Scientific). Blanks and a standard mix were processed following the same steps as the samples and used as quality controls. Derivatisation and gas chromatography coupled to mass spectrometry (GC-MS)-based metabolite profiling were conducted according to Fiehn et al. [115,116], with minor modifications as previously described for wheat seeds [55]. Vacuum-dried samples were resuspended in 10 μL of 20 mg/mL methoxyamine hydrochloride (Supelco 33045-U, Sigma-Aldrich, St. Louis, MO, USA) in pyridine (270970 Sigma Aldrich, St. Louis, MO, USA) and incubated at 600 rpm and 28 °C for 90 min under shaking in a thermomixer (Ditabis^®^ MHR 13, GML, Innsbruck, Austria). Then, 90 µL of *N*-methyl-*N*-trimethylsilyl-trifluoroacetamide (MTSTFA, 394866, Sigma-Aldrich, St. Louis, MO, USA) was added, and the derivatisation reaction continued for 30 min at 37 °C. After cooling at RT, 1 µL of each sample was injected in splitless mode into a Trace 1300 gas chromatograph coupled to a TSQ8000 triple quadrupole mass spectrometer (Thermo Fisher Scientific, Waltham, MA, USA) and equipped with a 30-m Rxi-5SilMS column with 0.25 µm cross bond 1,4-bis(dimethylsiloxy)phenylene dimethyl polysiloxane film and a 10-m integra-guard pre-column (13623-127 Restek Corporation, Bellefonte, PA, USA). The analysis and data processing followed the same workflow as previously reported [55], using the Automated Mass Spectral Deconvolution and Identification System (AMDIS v2.73; [117]) to first inspect the raw data and the Xcalibur^TM^ software (v4.0, Thermo Fisher Scientific, Waltham, MA, USA) for quantification. Relative values of LMW metabolite contents were obtained after normalising the peak areas of each metabolite to that of the internal ^13^C_6_-sorbitol standard and to the DW of individual samples.

### 4.5. HPLC Analysis of Low-Molecular-Weight Thiols and Disulphides

Dry seeds of the F_1_ (*n* = 5 replicates of ca. 13 mg seed FW pooled from individual plants of both lines) were frozen in liquid nitrogen, lyophilised, and ground as described for GC-MS analyses. gsh, cys, γ-glu-cys, cys-gly, and their corresponding disulphides were extracted on ice from 11.74 ± 0.67 mg of lyophilised and finely ground seed powder with 1 mL of 0.1 M HCl. Approximately 12 mg of polyvinylpolypirrolidone were also included before resuspension at 30 Hz for 30 s with a bead mill (Tissue Lyser II, Qiagen, Hilden, Germany). After a first centrifugation step (28,000× *g*, 20 min, 4 °C), a 700-µL aliquot of the supernatant was promptly transferred to a new Eppendorf tube and further centrifuged (28,000× *g*, 20 min, 4 °C). The assay is based on fluorescence labelling of thiol groups by monobromobimane (mBBr; [118]), occurring in the pH range between 8.0 and 8.3, which was achieved by combining the extracts with 200 mM bicine buffer at pH 8.8. A first aliquot of the extracts was used to label total thiols and disulphides after reducing disulphides with dithiothreitol (DTT, Carl Roth GmbH+Co, Karlsruhe, Germany). A second aliquot was combined with *N*-ethylmaleimide (NEM) to block free thiol groups and quantify disulphide concentrations only. Excess NEM and NEM-bound-thiols were removed by washing five times with toluene, and the remaining disulphides were reduced with DTT before labelling with mBBr. LMW thiols were separated by reversed-phase HPLC using a HPLC 1100 system (Agilent Technologies, Inc., Santa Clara, CA, USA) with a ChromBudget 120-5-C18 column (250 × 4.6 mm, 5.0 µm particle size, Bischoff GmbH, Leonberg, Germany) including a NUCLEODUR C18 Pyramid guard column (4 × 3 mm, 5.0 µm particle size, Macherey-Nagel GmbH, Düren, Germany) and detected by fluorescence (excitation: 380 nm; emission: 480 nm). The concentrations of LMW thiols and their corresponding disulphides were calculated using calibration curves of individual LMW thiol standards and subtracting the amounts of disulphides (in thiol equivalents) from the amounts of thiols plus disulphides, as described earlier [118].

The E_GSSG/2GSH_ was calculated according to the Nernst equation (Equation (1)). Seed WC was determined by measuring sample weights before and after seed lyophilisation, expressed as g H_2_O g^−1^ DW, and used to estimate the molar concentrations of GSH and GSSG according to Equation (1):(1)$$E_{GSSG/2GSH}=E^{0\prime }-\frac{R\;T}{n\;F}ln\frac{{\left[GSH\right]}^{2}}{\left[GSSG\right]}$$
whereby *R* is the gas constant (8.314 J K^−1^ mol^−1^); T, temperature in K; *n*, number of transferred electrons (2 GSH → GSSG + 2 H^+^ + 2 e^−^); *F*, Faraday constant (9.649 × 10^4^ C mol^−1^); *E*^0^’, standard half-cell reduction potential at cellular pH of 7.3 (*E*^0^’_GSSG/2GSH_ = −258 mV; [43,119]).

### 4.6. High Resolution Respirometry

Dry seeds of the F_2_ (*n* = 4 replicates of ca. 10 mg of both lines) were characterised for OCRs, using a FluoRespirometer Oxygraph-2k (Oroboros Instruments GmbH, Innsbruck, Austria). Before each measurement *via* Clark-type electrodes (Oroboros Instruments GmbH, Innsbruck, Austria), chambers were calibrated to 260.82–262.23 µM O_2_ and corrected for baseline signal with the DatLab software (Oroboros Instruments GmbH, Innsbruck, Austria). Dry seeds (9.98 ± 0.22 mg) were directly imbibed in the electrode chambers containing 2 mL of ultrapure H_2_O. Temperature was held constant at 21 °C, while stirring the seeds at 750 rpm in sealed electrode chambers. Data were acquired every 2 s with a polarisation voltage of 800 mV. Seed OCRs, indicative of mitochondrial respiration, were calculated during 15 min intervals in the plateau phase (always between 2 and 3 h after the onset of seed imbibition). Thereafter, 25 mM of KCN was added to block respiration by inhibiting complex IV. Finally, 25 µM of *n*-octyl-gallate, an inhibitor of the mitochondrial alternative oxidase, was injected to assess O_2_ consumption by this KCN-insensitive bypass of the mitochondrial ETC [72,120]. Seed OCRs were normalised on a DW basis after seed lyophilisation for 5 d. 

### 4.7. Bioinformatics and Statistical Analyses

Multiple sequence alignments were conducted using Clustal Omega (European Molecular Biology Laboratory and European Bioinformatics Institute; [121]).

Data were tested for significance at α = 0.05 by either one-way ANOVA analyses followed by Tukey’s Honest Significant Difference (HSD) test for *post-hoc* multiple comparisons of means, or Student’s *t*-test for comparing variables between the WT and mutant lines. All data were analyses using the software package SPSS Statistics 25 (IBM, New York, NY, USA). In each dataset, values with a Cook’s distance larger than 4/*n* (whereby *n* was the total number of observations in a dataset) were disregarded. The assumptions of normal distribution and homoscedasticity of variances were assessed via the Shapiro-Wilk and Levene’s tests, respectively. Whenever the assumption of normality was not fulfilled, data underwent non-parametric analyses (Kruskal-Wallis rank variance and Mann-Whitney *U*), followed by Bonferroni correction for multiple tests. For data showing heteroscedastic variances, simple bias-corrected accelerated bootstrap analyses were run with a sample size of 10^5^ and two different seeds (i.e., 5000 and 500), using the Mersenne Twister random number generator. The 95% confidence intervals of the bootstrap analyses’ outputs were sensitive to the seed size at the first decimal digit. Probit regression was run to interpolate (final TG > 50%) or extrapolate (final TG < 50%) the P50 values from survival curves [36]. P50 values were fitted between 100 and 0% TG with logistic regression and Levenberg-Marquardt iteration algorithm with the software package Origin^®^ 2017 (OriginLab Corporation, Northampton, MA, USA).

## 5. Conclusions

In conclusion, our data support that the lack of AtFAHD1a activity during seed development can have downstream consequences on seed quality traits. Therefore, we propose AtFAHD1a as a novel candidate player in seed metabolism, which influences cellular redox poise, seed thermo-dormancy, and longevity.

This study focused on the characterisation of *Atfahd1a-1* seed phenotype. The metabolic profile of dry seeds revealed that *AtFAHD1a* knockout influenced metabolic rearrangements typical of the late seed maturation and desiccation, in line with highest expression levels of *AtFAHD1a* during these stages of plant life cycle. Nonetheless, further research is required to reconcile influences on primary metabolism and the *Atfahd1a-1* seed phenotype. Considering the low constitutive expression of *ATFAHD1b,* an *ATFAHD1a* paralogue sharing high sequence homology also with hFAHD1, future studies to elucidate the functional role of AtFAHD1-containg proteins may benefit from a more extensive seed phenotypic characterisation, including *ATFAHD1b* knockout lines and *AtFAHD1a AtFAHD1b* double mutants. Additionally, a full validation of the biochemical profiles of *Atfahd1a-1* and WT seeds would also require accurate determination of AtFAHD1a and AtFAHD1b enzymatic activities, which may also contribute to ascertain the biochemical function of these FAHD-containing proteins in *A. thaliana* and generally in plant metabolism. 

## Figures and Tables

**Figure 1 ijms-22-02997-f001:**
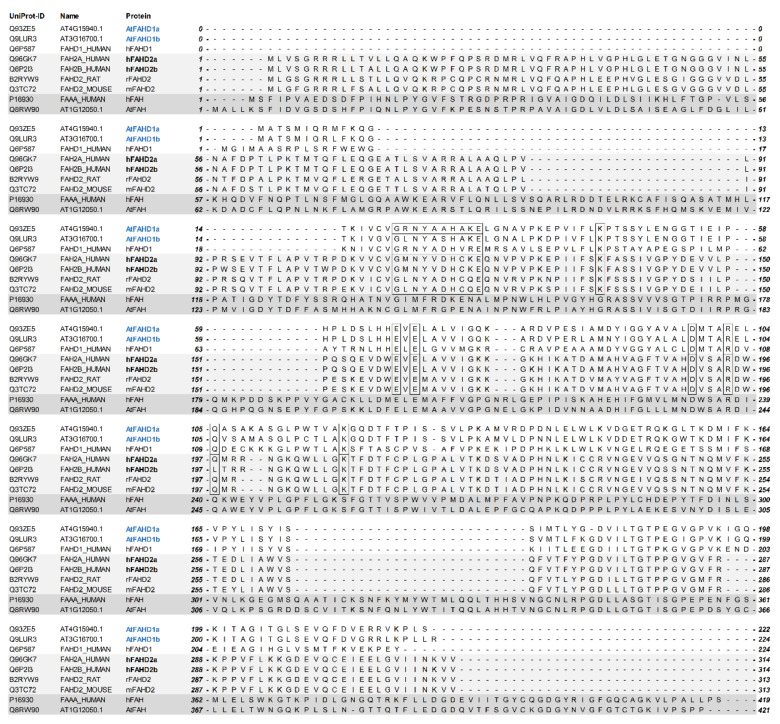
Multiple sequence alignment of the three *Arabidopsis thaliana* (At) fumarylacetoacetate hydrolase domain (FAHD)-containing proteins with orthologues. White and light grey shadings indicate the highest homology of AT4G15940.1 and AT3G16700.1 with human FAHD-containing protein 1 (hFAHD1), and they all differ from FAHD2 proteins in the N-terminus. AT1G12050.1 is orthologue to human fumarylacetoacetate hydrolase (hFAH) and was already identified as AtFAH [18]. Black borders refer to amino acids critical for the catalytic reaction of hFAHD1, which were fully conserved in AT4G15940.1 and AT3G16700.1. Their BLASTp alignment with hFAHD1 revealed sequence identities of 53 and 49%, respectively. Therefore, similar to the nomenclature assigned to hFAHD2a and hFAHD2b in human, AT4G15940.1 and AT3G16700.1 were renamed as AtFAHD1a and AtFAHD1b, respectively.

**Figure 2 ijms-22-02997-f002:**
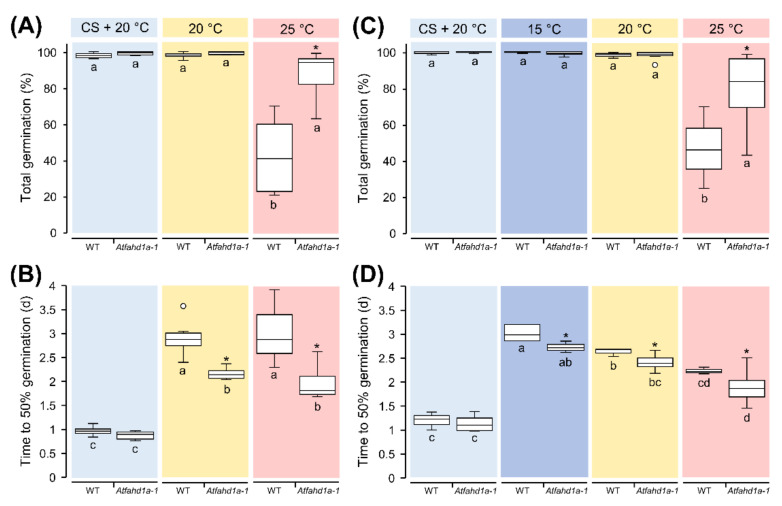
Germination phenotypes of wild type seeds [WT, ecotype Columbia-0 (Col-0)] and *Atfahd1a-1* (fumarylacetoacetate hydrolase domain-containing protein 1a) T-DNA insertional mutant seeds in the Col-0 background. To reveal differences in thermo-dormancy, seeds were cold stratified (CS) prior to germination at 20 °C and directly germinated at the indicated temperatures. (**A**,**B**) Total germination (TG) and time to 50% germination (T50) of WT and *Atfahd1a-1* seeds produced from the first plant generation. (**C**,**D**) TG and T50 of WT and *Atfahd1a-1* seeds produced from the second plant generation. Data are shown as box-plots (at 25 °C, *n* = 10 and 13 replicates of 100 seeds for WT and *Atfahd1a-1* lines, respectively; for all other conditions, *n* = 6 replicates of 100 seeds of both lines) and outliers as white circles. Within each panel, data labelled with the same letter do not differ significantly (one-way ANOVA followed by *post-hoc* Tukey’s HSD test, α = 0.05) and asterisks denote significant differences between WT and *Atfahd1a-1* seeds germinated at the same temperature (*t*-tests, *p* < 0.05).

**Figure 3 ijms-22-02997-f003:**
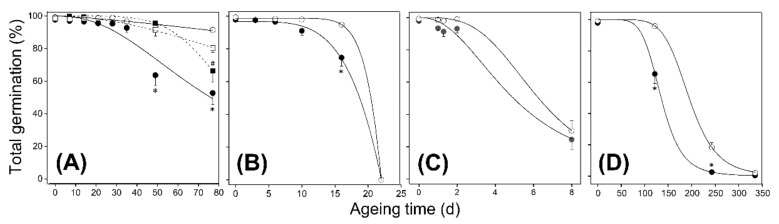
Survival curves of wild type seeds (WT, ecotype Columbia-0 [Col-0], closed symbols) and *Atfahd1a-1* (fumarylacetoacetate hydrolase domain-containing protein 1a) T-DNA insertional mutant seeds in the Col-0 background exposed to different regimes of controlled deterioration (CD) and accelerated ageing (AA) at 40 °C, as well as to ageing under ambient temperatures. At indicated intervals, seed viability was assessed by scoring total germination after 14 d at 20 °C. Circles and squares indicate seeds of the first (F_1_) and second (F_2_) plant generation, respectively. Open and closed symbols refer to WT and *Atfahd1a-1* seeds, respectively. (**A**) Seeds exposed to CD at 59.4 ± 1.1% RH. (**B**) Seeds exposed to CD at 74 ± 0.5% RH. (**C**) Seeds exposed to AA at 100% RH. (**D**) Seeds aged under ambient temperature and 68.2 ± 2.0% RH. Data refer to means ± SE (*n* = 6 replicates of 100 seeds for each line). Curves were drawn after fitting to the logistic function. Asterisks (for F_1_) and hashes (for F_2_) indicate significant differences between WT and *Atfahd1a-1* at the same ageing intervals (Mann-Whitney *U* tests, *p* < 0.05). Error bars within symbols are not shown.

**Figure 4 ijms-22-02997-f004:**
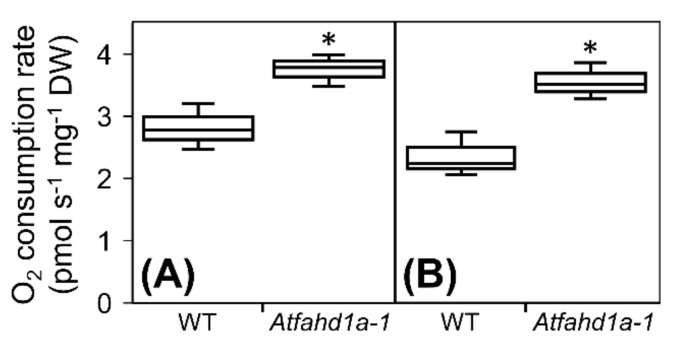
Oxygen (O_2_) consumption rate of wild type seeds (WT, ecotype Columbia-0 [Col-0]) and *Atfahd1a-1* (fumarylacetoacetate hydrolase domain-containing protein 1a) T-DNA insertional mutant seeds in the Col-0 background during the first 2 h after the onset of imbibition at 21 °C. (**A**) Maximal respiration rate corrected for basal respiration. (**B**) Activity of mitochondrial complex IV calculated by subtracting from corrected maximal respiration the respiration measured after exposing seeds to 25 mM KCN. Data are shown as box-plots (*n* = 4 replicates of ca. 10 mg of seeds of both lines) and were obtained from the second plant generation. Results were normalised on a dry weight (DW) basis. Asterisks denote significant differences between WT and *Atfahd1a-1* seeds (*t*-tests, *p* < 0.05).

**Figure 5 ijms-22-02997-f005:**
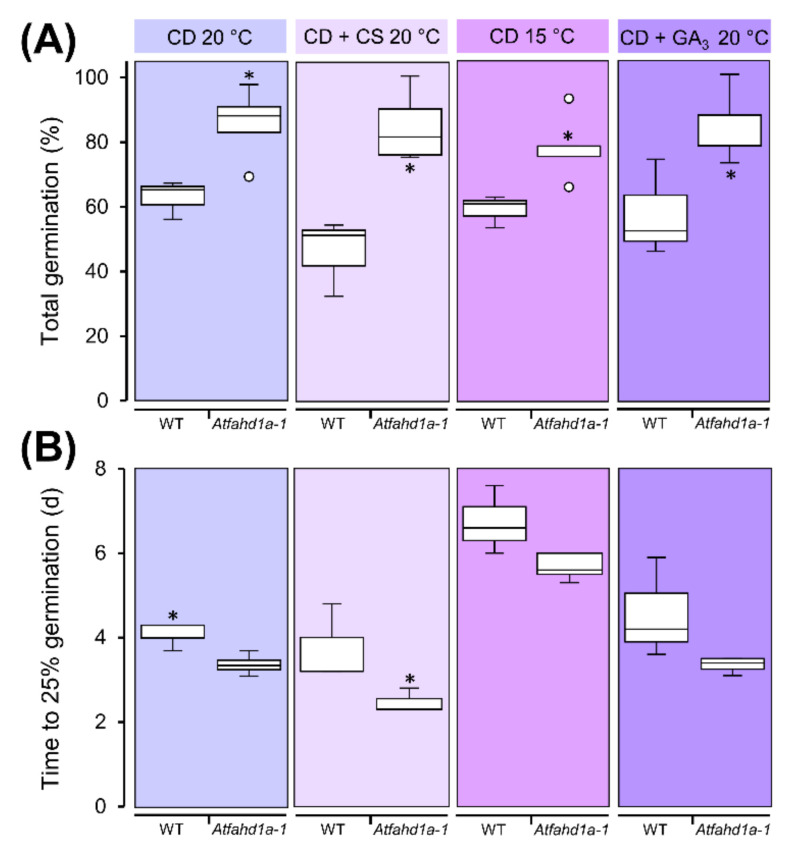
Longevity phenotypes of the second generation of wild type seeds [WT, ecotype Columbia-0 (Col-0)] and *Atfahd1a-1* (fumarylacetoacetate hydrolase domain-containing protein 1a) T-DNA insertional mutant seeds in the Col-0 background after 77 d of controlled deterioration (CD) at 40 °C and 59.4 ± 1.1% RH. After CD, seeds were directly incubated at 20 °C or exposed to dormancy breaking treatments prior to germination at 20 °C. These included cold stratification (CS) and imbibition with 1 µM of gibberellic acid (GA_3_). To assess the presence of residual thermo-dormancy after CD, seeds were also incubated at 15 °C. (**A**) Seed viability measured as total germination. (**B**) Germination rates expressed as time to 25% germination. Data for the various germination conditions are shown as box-plots (*n* = 6 replicates of 100 seeds of both lines) and outliers as white circles. Asterisks denote significant differences between WT and *Atfahd1a-1* seeds (Mann-Whitney *U* tests, *p* < 0.05).

**Figure 6 ijms-22-02997-f006:**
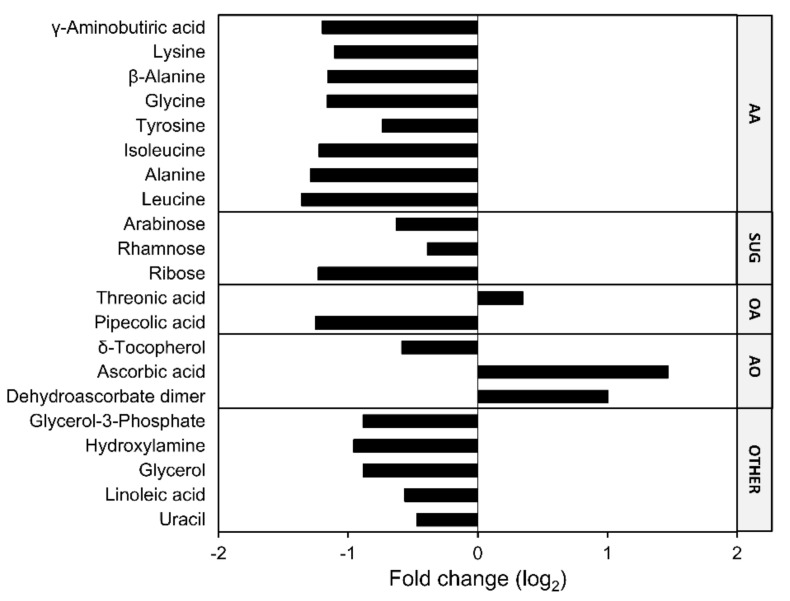
Low-molecular-weight metabolites with differential changes (*p* < 0.05, *t*-tests) in dry seeds of an insertional mutant line (fumarylacetoacetate hydrolase domain-containing protein 1a, *Atfahd1a-1*) in relation to the wild type (WT, ecotype Columbia-0). Average metabolite concentrations detected in *Atfahd1a-1* seeds were divided by those measured in the WT, and fold-changes of individual metabolites were expressed as log_2_. Data are means (*n* = 4) of individual metabolites detected by gas chromatography coupled to mass-spectrometry-based metabolite profiling of seeds collected from individual plants of the first plant generation for each line. Metabolites are grouped in biochemical classes: AA, amino acids; SUG, sugars; OA, organic acids; AO, antioxidants. All identified metabolites showing non-significant changes between the two seed lines (*p* ≥ 0.05, *t*-tests) are listed in Appendix A.

**Figure 7 ijms-22-02997-f007:**
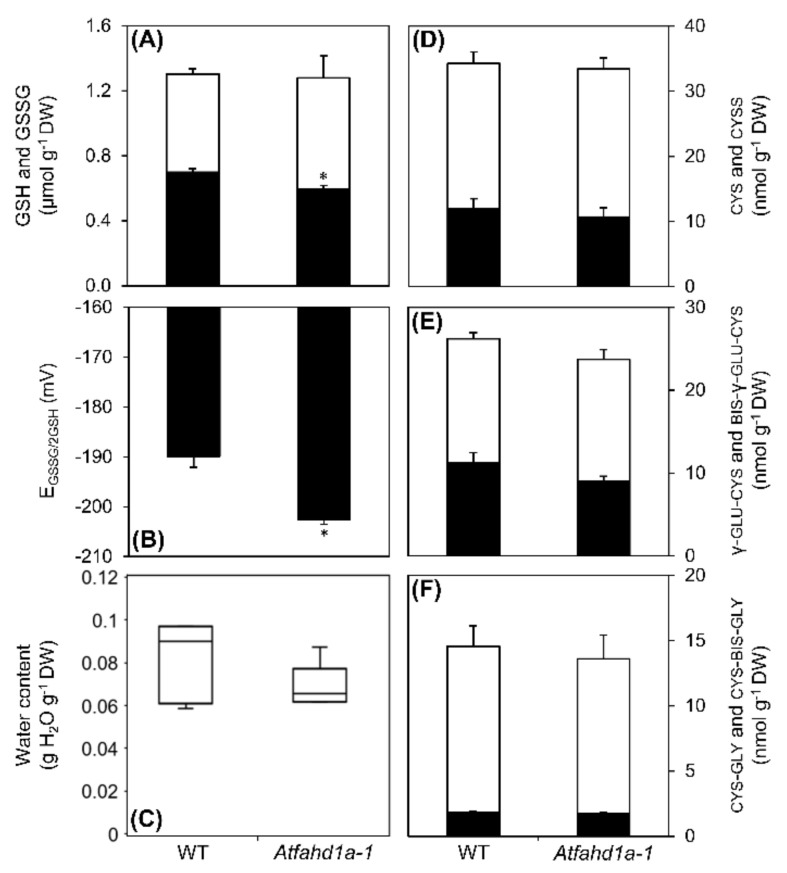
Changes in the concentrations of low-molecular-weight thiols (white bars) and corresponding disulphides (black bars) in wild type seeds (WT, ecotype Columbia-0 [Col-0]) and *Atfahd1a-1* (fumarylacetoacetate hydrolase domain-containing protein 1) T-DNA insertional mutant seeds in the Col-0 background. (**A**) Glutathione (GSH) and glutathione disulphide (GSSG). (**B**) Half-cell reduction potential of the GSSG/2GSH redox couple (E_GSSG/2GSH_) calculated using molar concentrations, which were estimated through the seed water contents shown in (**C**). (**C**) Water contents of dry seeds equilibrated to 36-38% RH, prior to the analytical measurements shown in the other panels. (**D**) cysteine (cys) and cystine (cyss). (**E**) γ-l-glutamyl-cysteine (γ-glu-cys) and bis-γ-glutamyl-cystine (bis-γ-glu-cys). (**F**) cysteinyl-glycine (cys-gly) and cystinyl-bis-glycine (cys-bis-gly). Data refer to seeds pooled from individual plants of the first generation for each line and are means ± SE (*n* = 5 replicates for both lines) with asterisks indicating significant differences (*t*-test, *p* < 0.05).

## Data Availability

Microarray data presented in this study are openly available in the Arabidopsis eFP Browser https://bar.utoronto.ca/efp/cgi-bin/efpWeb.cgi accessed on 15 January 2021.

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
