# Peer review of "AtFAHD1a: A New Player Influencing Seed Longevity and Dormancy in Arabidopsis?"

_ijms, 2021, doi:10.3390/ijms22062997_

Round 1
Reviewer 1 Report
The manuscript by Gerna et al. describe the function of a previously not characterized FAHD1a gene in seed dormancy and seed longevity of Arabidopsis. By using metabolite analyses they came up with its role in redox regulation to explain the phenotype of the knock -out mutant. All experiments are well described and [properly analyzed. However this reviewer do not understand the generation nomenclature they use for their materials. In material and methods the F2 seeds are mentioned as being grown in the same conditions as the F1 seeds. Nowhere is the origin of the F1 seeds mentioned. F1 seeds are obtained from crossing two genotypes. Nothing is mention on which parents were crossed. The authors mention that only homozygous T-DNA mutants were used for the experiments. Are these the F2 seeds? Heterozygosity is not unusual when the seeds from T-DNA mutants are obtained from the stock center and are still segregating for the T-DNA insertion However these heterozygous seeds are usually not described as F1 seeds. Please make this clear.
The authors mention that there Is a FAHD1b paralogue of the gene studied. I am surprised that the knock-out of this gene was not studied and even better why not make a double mutant and check the phenotypes of such double-mutants which may lead to stronger phenotypes as compared to the ones described for the FADH1a mutant. At least one would expect a discussion of the role of this gene in the discussion of the paper.
Reviewer 2 Report
The Authors « AtFAHD1a: a new player influencing seed longevity and dormancy in Arabidopsis? », provided a very complete analysis of the impact of an FAHD isoform on the seed germination vigour by studying seed longevity and dormancy. The results are very promising and support the conclusions of the Authors.
The ms is very well prepared and in particular I really appreciate the Discussion section.
I have only very minor remarks and questions:
Remarks:
- Contrary to “Arabidopsis”, “Arabidopsis thaliana” have to be written in italic.
- Please use small capital letter for l-glutamyl… and other related metabolites.
- A short conclusion could be interesting.
Question:
- The biochemical characterization of FAHD is still hypothetical in plants, even if your sequence analysis seems to indicate a conversed function as compared to hFAHD1. I am not a specialist of this enzyme, but it seems that that your metabolic analysis also provide some indications that support this hypothesis. Am I wrong or would you say this hypothesis would still be too risky in the current state?
